# Transcriptome and Metabolome Provide Insights into Fruit Ripening of Cherry Tomato (*Solanum lycopersicum* var. *cerasiforme*)

**DOI:** 10.3390/plants12193505

**Published:** 2023-10-09

**Authors:** Feng Pan, Qianrong Zhang, Haisheng Zhu, Junming Li, Qingfang Wen

**Affiliations:** 1Fujian Key Laboratory of Vegetable Genetics and Breeding, Crops Research Institute, Fujian Academy of Agricultural Sciences, Fuzhou 350002, China; 2State Key Laboratory of Vegetable Biobreeding, Institute of Vegetables and Flowers, Chinese Academy of Agricultural Sciences, Beijing 100081, China

**Keywords:** omics, cherry tomato, flavor, breeding

## Abstract

Insights into flavor formation during fruit ripening can guide the development of breeding strategies that balance consumer and producer needs. Cherry tomatoes possess a distinctive taste, yet research on quality formation is limited. Here, metabolomic and transcriptomic analyses were conducted on different ripening stages. The results revealed differentially accumulated metabolites during fruit ripening, providing candidate metabolites related to flavor. Interestingly, several key flavor-related metabolites already reached a steady level at the mature green stage. Transcriptomic analysis revealed that the expression levels of the majority of genes tended to stabilize after the pink stage. Enrichment analysis demonstrated that changes in metabolic and biosynthetic pathways were evident throughout the entire process of fruit ripening. Compared to disease resistance and fruit color genes, genes related to flavor and firmness may have a broader impact on the accumulation of metabolites. Furthermore, we discovered the interconversion patterns between glutamic acid and glutamine, as well as the biosynthesis patterns of flavonoids. These findings contribute to our understanding of fruit quality formation mechanisms and support breeding programs aimed at improving fruit quality traits.

## 1. Introduction

Improving fruit flavor while maintaining high yields, disease resistance, and long postharvest shelf life poses a significant challenge for breeders [1,2]. To achieve this goal, it is essential to gain a comprehensive understanding of how flavor compounds are formed throughout the fruit ripening process. Tomato (*Solanum lycopersicum*) serves as a primary model system for studying climacteric fleshy fruit development and ripening [3]. Cherry tomatoes (*Solanum lycopersicum* var. *cerasiforme*), as one of the tomato subspecies, are widely planted in China and are favored by consumers due to their intense flavor. Along with apples, grapes, and bananas, cherry tomatoes are among the four priority fruits supported by the Food and Agriculture Organization of the United Nations. Compared to the “big-fruit” tomato, cherry tomatoes exhibit higher levels of sugar (~44% higher), organic acids (~5% higher), and volatile organic compounds [4,5,6]. This makes them an excellent model for analyzing the formation of flavors.

The ripening process of red-fruited-clade tomato fruit can be divided into five stages: the mature green stage, the breaker stage, the pink stage, the light-red stage, and the red ripe stage. These stages are characterized by physiological and biochemical changes that ultimately lead to alterations in the appearance, texture, flavor, and aroma of the fruit [3,7]. Key manifestations of this process include changes in ethylene levels, gradual alterations in fruit color, fruit softening, and modifications in the metabolism of sugars, acids, and other compounds, as well as the production of volatile substances [8]. Specifically, the ripening process involves the accumulation and spread of autocatalytic ethylene, which leads to a series of phenotypic changes [9]. The accumulation of carotenoids, such as lycopene and β-carotene, contributes significantly to changes in color [3]. Fruit softening and increased juiciness are primarily caused by the partial disassembly of the cell wall [10]. Additionally, the formation of sugars, organic acids, and volatiles, and even some bitter compounds, such as steroidal alkaloids and their glycosylated forms, as well as flavanone derivatives, can affect the fruit’s flavor and aroma [11,12].

To date, numerous key genes related to fruit ripening in tomatoes have been identified, including *ACC*, *ACS*, and *ACO* genes involved in direct ethylene synthesis [13,14,15,16,17], *CNR* [18], *PSY1* [19,20], and *OG* [21] influencing fruit color transformation, *PG* [22,23,24], *PL* [25,26], and *PME* [27] affecting fruit softening and juice viscosity, as well as *E8* [28], *LIN5* [29], *PAR1*, and *PAR2* [30] influencing fruit flavor. Furthermore, various metabolites that impact fruit flavor formation have been documented, encompassing sugars, acids, amino acids, fatty acids, terpenoids, bitter metabolites, and other substances [12]. However, despite these advancements, connecting multiple ripening-related genes with the signaling and coordination mechanisms that induce ripening remains a significant challenge [31]. The complex gene expression networks associated with metabolite changes during fruit ripening largely remain unknown.

Omics approaches, such as metabolomics and transcriptomics, can contribute to our understanding of the genetic, hormonal, and metabolic networks that regulate tomato fruit development and ripening [32,33]. Integrative analysis of metabolite and transcript levels is a well-established approach that has been applied to various fruit crops including tomato [34], ponkan [35], kiwifruit [36], and litchi [37]. However, cherry tomatoes, being a popular commodity in the fresh market, have been relatively under-studied in this context. Hence, in this study, we conducted a combined metabolomic and transcriptomic analysis of cherry tomatoes to investigate the mechanisms underlying flavor formation during ripening. We selected the Yu-nu cultivar as the material due to its widespread cultivation in southeast China, as well as its reputation for high yield, medium firmness, and intense flavor. Considering the growth pattern and flavor changes, fruit samples at the mature green (MG), pink (Pk), and red ripe (RR) stages were collected for analysis (Figure 1a). Through the identification of differentially accumulated metabolites (DAMs) and differentially expressed genes (DEGs), we explored the changes in metabolite accumulation and gene expression during fruit ripening. The integrated analysis provided a systematic perspective on the biosynthesis of flavor-related metabolites. These findings contribute to our understanding of tomato breeding strategies aimed at striking a balance between consumer sensory experience and expected production qualities.

## 2. Results

### 2.1. Metabolic Changes during Fruit Ripening

In order to investigate the metabolic changes that occur during fruit ripening, we conducted a quantification of metabolites at three different stages of ripening: mature green (MG), pink (Pk), and red ripe (RR). A representative picture of the fruits is shown in Figure 1a. Principal component analysis (PCA) based on raw mass spectrometry data revealed similarities among replications and noticeable differences between the three ripening stages (Appendix A and Figure 1b). Overall, we identified a total of 420 metabolites, categorized into various groups including 67 benzenoids, 66 organic acids, 62 fatty acyls, 27 polyketides, 25 nucleotides and nucleotide derivatives, and 16 carbohydrates and other metabolites (Appendix A, Appendix A).

During the MG-Pk stage, a total of 71 differentially accumulated metabolites (DAMs) were detected (Figure 1b), with 50 metabolites showing an increase and 21 metabolites showing a decrease (Figure 1c). The increased metabolites were found to be significantly enriched in aromatic metabolite sets, such as benzamides, purines, pyrimidines, cinnamic acids, and benzenediols, as well as sugar-related metabolite sets like disaccharides and glycosyl compounds. On the other hand, the decreased metabolites were significantly enriched in unpleasant-smelling pyridines, monosaccharides, and TCA acids. These changes in metabolite levels may have an impact on the initial formation of flavor and aroma.

Similarly, during the Pk-RR stage, a total of 79 DAMs were detected (Figure 1b), with 39 metabolites showing an increase and 40 metabolites showing a decrease (Figure 1d). The increased metabolites were enriched in aromatic metabolite sets such as benzamides, purines, phenylacetic acids, and phenols, as well as in the odor-related metabolite set of aldehydes. Likewise, the decreased metabolites were also enriched in aromatic metabolite sets including benzamides, indoles, cinnamic acids, and benzenediols. These metabolites may serve as intermediates in the process of flavor and aroma formation.

Overall, across all three stages, a total of 24 DAMs were detected (Figure 1b). Among these, 12 metabolites (2-hydroxyglutarate, 2-isopropylmalic acid, adenosine, coumarin, galactaric acid, L-aspartic acid, L-glutamic acid, L-proline, p-coumaroyl quinic acid, phosphorylcholine, psilocybin, and salidroside) showed continuous increments, 4 metabolites (5-methyltetrahydrofolic acid, D-ribose, L-threonine, and orotic acid) showed continuous decrements, and 10 metabolites (12-hydroxydihydrochelirubine, 2-dehydro-3-deoxy-L-rhamnonate, D-xylitol, glycerophosphocholine, L-homophenylalanine, m-coumaric acid, putrescine, pyroglutamic acid, quercetin 3-2G-xylosylrutinoside, and trehalose) exhibited an initial increase followed by a decrease (Figure 1b, Appendix A). These DAMs might provide candidate metabolites related to flavor.

### 2.2. Gene Expression Changes during Fruit Ripening

To investigate the expression patterns of genes during fruit ripening, we conducted whole-transcriptome sequencing at three distinct ripening stages. A total of 47.85 Gb of raw sequences with a read length of 150 bp was generated. From this dataset, we retained 68,677,640 high-quality reads, achieving an average alignment rate of 94.8% (Appendix A). Utilizing the RNA-seq analysis pipeline, we detected 16,978 genes expressed during fruit ripening, with an average read count per million (CPM) equal to or greater than 1. Notably, 13,514 genes were found to be expressed across all three stages.

A total of 2169 differentially expressed genes (DEGs) were identified in fruits between the MG stage and Pk stage, including 680 upregulated genes and 1489 downregulated genes (Appendix A, Figure 2a). Gene Ontology (GO) annotations of the DEGs revealed enrichment in biological processes such as organ development and morphogenesis, cellular components including the membrane and cell wall, and molecular functions like catalytic activity and transporter activity (Appendix A, Figure 2b). Furthermore, Kyoto Encyclopedia of Genes and Genomes (KEGG) annotations of the DEGs indicated enrichment in metabolism, flavonoid biosynthesis, isoflavonoid biosynthesis, phenylpropanoid biosynthesis, as well as in the biosynthesis of other secondary metabolites (Appendix A, Figure 2c).

Subsequently, in fruits transitioning from the pink (Pk) stage to the ripe red (RR) stage, we identified a total of 1165 DEGs. Among these DEGs, 165 genes were upregulated, while 1000 genes were downregulated (Appendix A, Figure 2d). The GO annotations of the DEGs revealed enrichment in the biological processes of metabolism and photosynthesis, the cellular components of the extracellular region, cell wall, and plastid, and the molecular functions of catalytic activity and lyase activity (Appendix A, Figure 2e). Moreover, the KEGG annotations of the DEGs indicated enrichment in metabolism, photosynthesis, and biosynthesis of other secondary metabolites (Appendix A, Figure 2f).

The comparison between the MG-Pk and Pk-RR stages revealed only 356 common DEGs, indicating distinct physiological and biochemical responses before and after the pink stage of fruit ripening. We focused on analyzing well-characterized genes involved in color, texture, and flavor from the DEGs (Figure 2g). Interestingly, most of the genes related to texture and flavor showed significant upregulation or downregulation at the Pk stage, and their expression levels remained stable during the RR stage. For instance, *AFF*, known for promoting locule gel formation and improving firmness and solids content at low expression levels [38], showed lower expression at the Pk stage and insignificant changes during the RR stage. Similarly, *PAR2*, which enhances the accumulation of 2-phenylethanol and reduces the accumulation of 2-phenylacetaldehyde, thereby influencing aroma and flavor [30], displayed a similar expression pattern to *AFF* at the Pk-RR stage. However, the expression of several genes related to flavor showed continuous changes from the Pk stage to the RR stage. For example, *ADH2*, which encodes alcohol dehydrogenase and affects the aroma derived from the lipoxygenase pathway [39], was continuously upregulated from the MG stage to the RR stage. This suggests that these genes might enhance flavor during the whole ripening stage.

There were 80 KEGG terms commonly enriched in both the MG-Pk and Pk-RR stages, mainly involving metabolite biosynthesis and various metabolic pathways. These findings reflect the continuous variations in abundant metabolites during the fruit ripening process (Appendix A, Appendix A). Furthermore, 30 KEGG terms were exclusively enriched in the MG-Pk stage, encompassing several pathways involved in signaling and transport. Additionally, 13 KEGG terms were exclusively enriched in the Pk-RR stage, incorporating pathways related to energy and photosynthesis, which may be associated with the physiological transition from chloroplasts to chromoplasts and seed development [40,41,42]. These enrichment analyses of DEGs reveal significant differences in metabolic pathways among the different ripening stages.

### 2.3. Correlation Analysis between DAMs and DEGs

To understand patterns linking the transcriptome and metabolome, correlations were calculated between the abundance of DEGs and DAMs. A total of 5332 pairwise correlations were identified, involving 1289 genes and 97 metabolites. Among these correlations, the top ten metabolites displayed the highest number of correlations with 934 genes (Figure 3a, Appendix A), including seven metabolites (adenosine, amygdalin, cytosine, naringenin, naringenin chalcone, ubiquinone-1, and umbelliferone) that were found to be negatively correlated with the expression of a majority of DEGs and three metabolites (spermine, carvone, and beta-alanyl-L-lysin) that were positively correlated, indicating their involvement in a complex regulatory network during fruit ripening.

Furthermore, to understand the correlation between gene expression and metabolite accumulation, we examined the DAMs that showed significant correlations with well-characterized genes related to color, texture, disease resistance, and flavor (Appendix A, Figure 3b). Interestingly, our analysis revealed that the genes involved in flavor exhibited more pronounced changes in metabolite abundance compared to those correlated with color (T-test *p*-value = 8.83 × 10^−3^). This observation suggests that alterations in fruit flavor are accompanied by a wide range of changes in metabolite composition, while fruit color may not exert a dominant influence on flavor. Additionally, none of the DAMs were significantly correlated with disease resistance genes, while two of eight firmness-related genes (*ERF.D7* and *TBG6*) were found to be correlated with five or more metabolites (Appendix A), suggesting that firmness-related genes might led to a greater change in metabolites compared to disease resistance genes.

### 2.4. Integrative Analysis of KEGG Pathways

To investigate the dynamic changes in fruit flavor during ripening, we performed an integrative analysis of the transcriptome and metabolome. Previous studies have demonstrated a significant association between glutamic acid and overall fruit flavor intensity and consumer preference [2]. Meanwhile, more DAMs and DEGs were identified in the glutamate metabolism pathway, facilitating us to gain insights into the metabolic transformations in this pathway (Appendix A). As depicted in Figure 4, the concentration of glutamic acid increased continuously throughout fruit ripening, concomitant with a gradual decrease in oxoglutaric acid levels, which is consistent with the flavor development process. Additionally, we observed the upregulation of *Solyc04g014510* (*gst1*, encoding glutamine synthetase cytosolic isozyme 1-1) and *Solyc11g011380* (*GS1*, encoding glutamine synthetase), along with the downregulation of *Solyc03g083440* (*LOC101254281*, encoding glutamate synthase 1), indicating a reciprocal conversion between glutamic acid and glutamine during fruit ripening. Specifically, during the MG-Pk stage, glutamic acid is converted to glutamine, while during the Pk-RR stage, glutamine is converted back to glutamic acid. Furthermore, the conversion from glutamic acid to gamma-aminobutyric acid and then to succinic acid semialdehyde was found to occur during the Pk-RR stage, and the corresponding genes, *Solyc11g011920* (*GAD2*, encoding glutamate decarboxylase isoform 2), *Solyc01g005000* (*GAD3*, encoding glutamate decarboxylase isoform 3), and *Solyc12g006470* (*GAME12*, encoding gamma aminobutyrate transaminase 2), showed a lower expression level at this stage.

In the context of the flavonoid biosynthesis pathway, we identified nine DEGs and nine DAMs, providing a comprehensive view of the changes in flavonoid metabolism (Figure 5, Appendix A). Specifically, the expression of *Solyc01g096670* (*LOC101246092*, encoding p-coumaroyl quinate) showed a negative correlation with 5-O-caffeoylshikimic acid accumulation and a positive correlation with cholorogenic acid. Interestingly, both metabolites interacted with caffeoyl-CoA and displayed opposite accumulation profiles, suggesting a potential competitive relationship involving *Solyc01g096670* in their biosynthesis. Moreover, when comparing the expression profiles of the DEGs *Solyc02g083860* (*F3H*, encoding flavanone 3-dioxygenase), *Solyc11g013110* (*LOC101249699*, encoding flavonol synthase), and *Solyc03g115220* (*F3*′*H*, encoding flavonoid 3′-monooxygenase) with the accumulation profile of the dihydrokaempferol-kaempferol-quercetin pathway, we observed a closer resemblance, indicating that this pathway predominantly operates during fruit ripening, rather than the dihydrokaempferol-taxifolin-quercetin pathway. These findings contribute to our understanding of the mechanisms underlying the formation of fruit quality.

## 3. Discussion

The formation of fruit flavor is a complex and multifactorial process that is not yet fully understood. In this study, we aimed to gain a comprehensive understanding of fruit ripening in cherry tomatoes by integrating transcriptome and metabolome data. By identifying DAMs and DEGs, we were able to unravel the patterns of metabolite accumulation and gene expression during fruit ripening. And the integrative analysis provided valuable insights into the development of improved fruit flavor and tomato breeding.

Fruit flavor development is typically associated with the ripening process [43]. Previous research has extensively focused on identifying metabolites related to tomato fruit flavor [30,33,44,45]. In our study, we detected a total of 124 DAMs in cherry tomatoes (cv. Yu-nu), some of which have been previously reported to be associated with flavor, such as glutamic acid and phenol [2,46]. Interestingly, certain key flavor compounds, including glucose, fructose, citric acid, and 2-phenylethanol, which have been reported to contribute to flavor preferences [1,30], did not show significant changes from the mature green (MG) to the fully ripe (RR) stage. We observed the expression levels of these traits-associated gene loci identified through GWAS [2]. Among the 41 genes, only 2 were found to be included in our identified DEG list, while the expression levels of the remaining genes did not show significant changes during the MG-RR stages (Appendix A). This suggests that these flavor-related metabolites may have already reached a stable state prior to the MG stage, ensuring the establishment of foundational flavor by the time of harvesting, even before reaching the RR stage. Successive detection of DEGs in the MG-Pk and Pk-RR stages resulted in the identification of 2169 and 1165 DEGs, respectively (Appendix A, Figure 2a,d). The decrease in DEG abundance suggests that gene expression tends to stabilize as the fruit ripens. Upregulation of DEGs involved in the ethylene synthesis pathway, such as *ACO1*, *ACO3*, *ACO5*, *ACS2*, and *ACS4*, promotes ethylene accumulation, leading to physiological and biochemical changes [17]. Moreover, the upregulated expression of *PSY1* [20] contributes to increased lycopene content in the fruit, resulting in a change in fruit color. Conversely, the continuous downregulation of *TGB6* [47] leads to reduced fruit firmness. Additionally, DEGs related to firmness, namely *FUL1* [48], *PG2a* [49], *qFIS1* [50], *SlERF.D7* [51], and *Solyc11g011300* [52], showed changed expression levels from the Pk stage onwards, indicating their role in affecting fruit softening beginning at the Pk stage. On the other hand, several DEGs, such as *ADH2* [53] and *CGT* [54], promoted flavor formation at both the MG-Pk and Pk-RR stages (Figure 2g). These expression patterns of genes related to firmness and flavor at different stages reflect the strategy through which cv. Yu-nu has gained recognition from both producers and consumers. The key flavor compounds are formed before the MG stage, and harvest before the pink stage ensures the firmness of the fruits during transportation. As the fruits are placed on the shelf, their texture gradually softens while other flavor compounds are enhanced, ultimately giving rise to the desirable flavor appreciated by consumers.

The Integrative analysis has deepened our understanding of the close relationship between gene expression and metabolite accumulation (Appendix A). One particular gene, *Solyc01g096670*, which encodes the p-coumaroyl quinate/shikimate 3′-hydroxylase, has been reported to be involved in the biosynthesis of both 5-O-caffeoylshikimic acid and chlorogenic acid [55]. It has been observed that the expression of *Solyc01g096670* promotes the accumulation of chlorogenic acid while suppressing the accumulation of 5-O-caffeoylshikimic acid during fruit ripening (Figure 5). Chlorogenic acid has been associated with a reduced risk of cancer and type 2 diabetes [56], while 5-O-caffeoylshikimic acid may ameliorate kidney injury caused by hyperuricemia [57]. These findings suggest that regulating the expression of *Solyc01g096670* could enhance the production of different organic acids that are beneficial to human health, offering insights into improving the nutritional quality of fruits.

The long-term improvements made in tomato fruit size, firmness, and disease resistance have led to a deterioration in fruit flavor [1,2]. It has been found that the selection for alleles of these genes has resulted in changes in metabolite profiles due to linkage with nearby genes [33]. Furthermore, research has indicated that shortening the length of disease resistance introgression segments improves other traits while retaining resistance [58]. In our study, nine disease resistance genes (*Cf-2* [59], *Cf-10* [60], *Ph-2* [61], *Rx4* [62], *Sm* [63], and four late-blight-resistant QTL [64]) were identified among the DEGs, while none of these genes were found to be significantly correlated with any metabolites, supporting that the deterioration in flavor might be caused by linkage segments near the resistance gene, rather than by the resistance gene itself. Meanwhile, correlation analysis indicated a stronger correlation between flavor-related genes and metabolites. Additionally, two firmness-related genes (*ERF.D7* and *TBG6*) were found to be correlated with five or more metabolites (Figure 3b, Appendix A). Therefore, our findings suggested that selection for firmness-related genes might have directly led to a greater deterioration in fruit flavor compared to selection for disease resistance genes, thus enhancing our understanding of the impact of the breeding improvement process on flavor.

## 4. Materials and Methods

### 4.1. Plant Materials

Cherry tomatoes (*Solanum lycopersicum* var. *cerasiforme* cv. Yu-nu) were cultivated in Pu-dang village, Xindian town, Fuzhou city, China, during the spring of 2020. The cultivation was carried out using soil-less techniques in greenhouses. Water-soluble fertilizers were applied, and the nutrient supply was closely integrated with the irrigation system. No supplemental lighting was utilized, relying solely on natural sunlight for plant growth. Samples were collected at three distinct stages: mature green (MG; fully sized, green fruit), pink (Pk; 50% pink color), and red ripe (RR; fully red). Immediately after harvesting, the fruits were frozen in liquid nitrogen for preservation. Biological triplicates were utilized for all experimental procedures.

### 4.2. Metabolome Analysis

Metabolites were extracted and analyzed using ultra-high-performance liquid chromatography–tandem mass spectrometry (UHPLC-MS/MS) at Biozeron Biotechnology Co., Ltd., located in Shanghai, China. In brief, the tissues were lyophilized and pulverized, and 200 mg frozen powder was weighed and extracted with 0.6 mL 2-Chloro-L-phenylalanine (4 ppm, formulated with methanol). After ultrasonic and centrifugal treatment, 300 μL supernatant was collected and filtered through a 0.22 μmol membrane filter. Then, the filtrate was transferred into the detection vial and used to perform LC-MS analysis. An amount of 20 µL filtrate from each tested sample was mixed to create a quality control (QC) sample, which was used to correct the deviation in the mixed sample analysis results and any errors caused by the analytical instrument itself. And LC-MS analysis was performed on the remaining tested samples

Chromatographic separation was accomplished in a Thermo Vanquish system equipped with an ACQUITY UPLC^®^ HSS T3 (150 × 2.1 mm, 1.8 µm, Waters) column maintained at 40 °C. The temperature of the autosampler was 8 °C. Gradient elution of analytes was carried out with 0.1% formic acid in water (A2) and 0.1% formic acid in acetonitrile (B2) or 5 mM ammonium formate in water (A3) and acetonitrile (B3) at a flow rate of 0.25 mL/min. Injection of 2 μL of each sample was performed after equilibration. An increasing linear gradient of solvent B (*v*/*v*) was used as follows: 0~1 min, 2% B2/B3; 1~9 min, 2~50% B2/B3; 9~12 min, 50~98% B2/B3; 12~13.5 min, 98% B2/B3; 13.5~14 min, 98~2% B2/B3; and 14~20 min, 2% B2-positive model (14~17 min, 2% B3-negative model).

The ESI-MSn experiments were executed on a Thermo Q Exactive Plus mass spectrometer with spray voltages of 3.8 kV and −2.5 kV in positive and negative modes, respectively. Sheath gas and auxiliary gas were set at 30 and 10 arbitrary units, respectively. The capillary temperature was 325 °C. The analyzer scanned over a mass range of *m*/*z* 81-1000 for full scan at a mass resolution of 70,000. Data-dependent acquisition (DDA) MS/MS experiments were performed with HCD scan. The normalized collision energy was 30 eV. Dynamic exclusion was implemented to remove some unnecessary information in the MS/MS spectra.

Data processing and metabolite identification were performed using MetaboAnalyst 5.0 (https://www.metaboanalyst.ca/, accessed on 5 July 2023). Differentially accumulated metabolites (DAMs) were identified using a combination of the t-test and the orthogonal projection to latent structures–discriminant analysis (OPLS-DA) model. Metabolites with a *p*-value less than 0.05 and a variable influence on projection (VIP) value greater than or equal to 1 were considered as DAMs. The OPLS-DA model analysis (Appendix A) was performed using the ropls package (version: 1.30.0) [65]. Metabolites were annotated with KEGG using KEGG Mapper (https://www.genome.jp/kegg/mapper/, accessed on 10 July 2023).

### 4.3. RNA Sequencing and Transcriptome Analysis

Total RNA was extracted from the 500 mg frozen powder using TIANGEN^®^ RNAprep Pure Plant Kit—DP441- (Tiangen Co., Ltd., Beijing, China) according to the manufacturer’s instructions and genomic DNA was removed using DNase I (Takara Bio Inc., Beijing, China). Then, RNA quality was determined using 2100 Bioanalyser (Agilent Co., Ltd., Beijing, China) and quantified using ND-2000 (Thermo Fisher Scientific Inc., Shanghai, China). High-quality RNA samples (OD260/280 = 1.8~2.2, OD260/230 ≥ 2.0, RIN ≥ 6.5, 28S:18S ≥ 1.0, >10 μg) were used to construct the sequencing library.

RNA-seq transcriptome libraries were prepared following the TruSeqTM RNA sample preparation Kit from Illumina (San Diego, CA, USA), using 1 μg of total RNA. Briefly, messenger RNA was isolated with polyA selection using oligo(dT) beads and fragmented using fragmentation buffer. cDNA synthesis, end repair, A-base addition, and ligation of the Illumina-indexed adaptors were performed according to Illumina’s protocol. Libraries were then size-selected for cDNA target fragments of 200–300 bp on 2% Low Range Ultra Agarose followed by PCR amplified using Phusion DNA polymerase (NEB-china, Beijing, China) for 15 PCR cycles. After quantification via TBS380, paired-end libraries were sequenced with Illumina NovaSeq 6000 sequencing (BIOZERON Co., Ltd., Shanghai, China).

Quality control measures were applied to the sequencing data using fastp software (version: 0.21.0) [66]. Alignment of the sequenced reads to the reference genome (version SL4.0) was carried out using Hisat2 software (version: 2.2.1) [67]. Subsequently, StringTie software (version: 2.2.1) [68] was used to assemble the aligned reads into potential transcripts, with default parameters.

Differential expression analysis was performed using the R package edgeR (version: 3.40.2) [69]. Genes showing significant differential expression were selected based on a log2 fold change of ≥2 and a Q value greater than 0.01.

For Gene Ontology (GO) and Kyoto Encyclopedia of Genes and Genomes (KEGG) enrichment analyses, the TBtools software (version v1.1.20) was employed [70].

### 4.4. Integrative Analysis of Metabolome and Transcriptome

In the subsequent analysis, an integrative approach was employed to explore the correlation between metabolite accumulation and gene expression.

Correlation analysis between metabolite accumulation and gene expression was performed using R software (version: 4.2.2). The significance of correlations was determined by setting a threshold with an absolute value of correlation coefficient greater than 0.95 and a *p*-value less than 0.01. This analysis aimed to identify highly systematic changes occurring during the ripening stage.

## 5. Conclusions

Tomato fruit ripening is a complex and systematic process, accompanied by the cessation of fruit development and abundant changes in metabolites. In this study, we identified differentially accumulated metabolites during the fruit ripening process, providing candidate metabolites related to flavor. Notably, key flavor-related metabolites already reached a stable level prior to the MG stage. Identification of differentially expressed genes revealed that the expression levels of the majority of genes tended to stabilize after the pink stage, and changes in metabolic and biosynthetic pathways occurred throughout the entire process of fruit ripening. Compared to disease resistance and fruit color genes, genes related to flavor and firmness might have a broader impact on the accumulation of metabolites. Additionally, we discovered interconversion patterns between glutamic acid and glutamine, as well as the biosynthesis patterns of flavonoids. These findings offer a systematic perspective on the accumulation of metabolites and changes in gene expression during the ripening of cherry tomato cv. Yu-nu. This study provides insights into the understanding of flavor formation mechanisms and can contribute to future breeding programs in cherry tomatoes, ultimately enhancing their flavor characteristics and quality.

## Figures and Tables

**Figure 1 plants-12-03505-f001:**
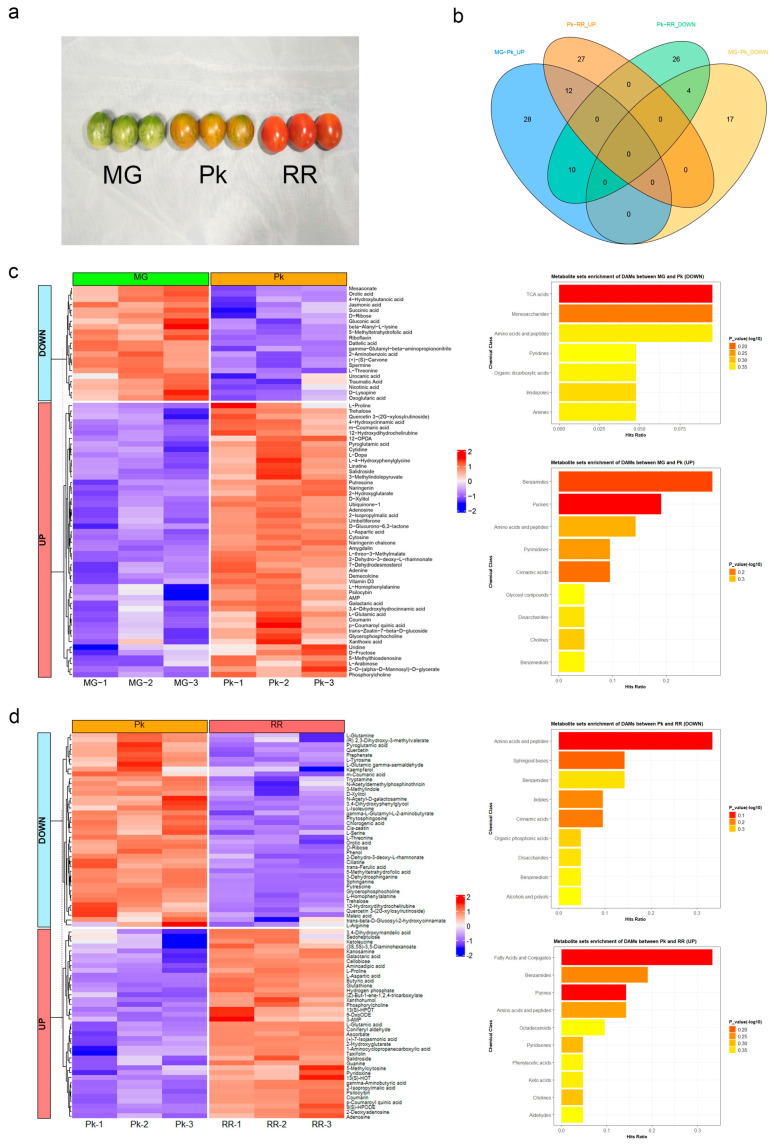
Metabolome analysis of fruit at different stages: mature green (MG), pink (Pk), and red ripe (RR): (**a**) Representative picture of Yu-nu fruit at three sampling stages. (**b**) Venn diagram showing differentially accumulated metabolites (DAMs) between MG-Pk and Pk-RR stages. (**c**) Accumulation profiles of DAMs from MG to Pk. The heatmap on the left illustrates changes in DAM accumulation, and the bar charts on the right depict the enrichment of metabolite sets for increased (UP) and decreased (DOWN) DAMs. (**d**) Accumulation profiles of DAMs from Pk to RR. The heatmap on the left illustrates changes in DAM accumulation, and the bar charts on the right depict the enrichment of metabolite sets for increased (UP) and decreased (DOWN) DAMs.

**Figure 2 plants-12-03505-f002:**
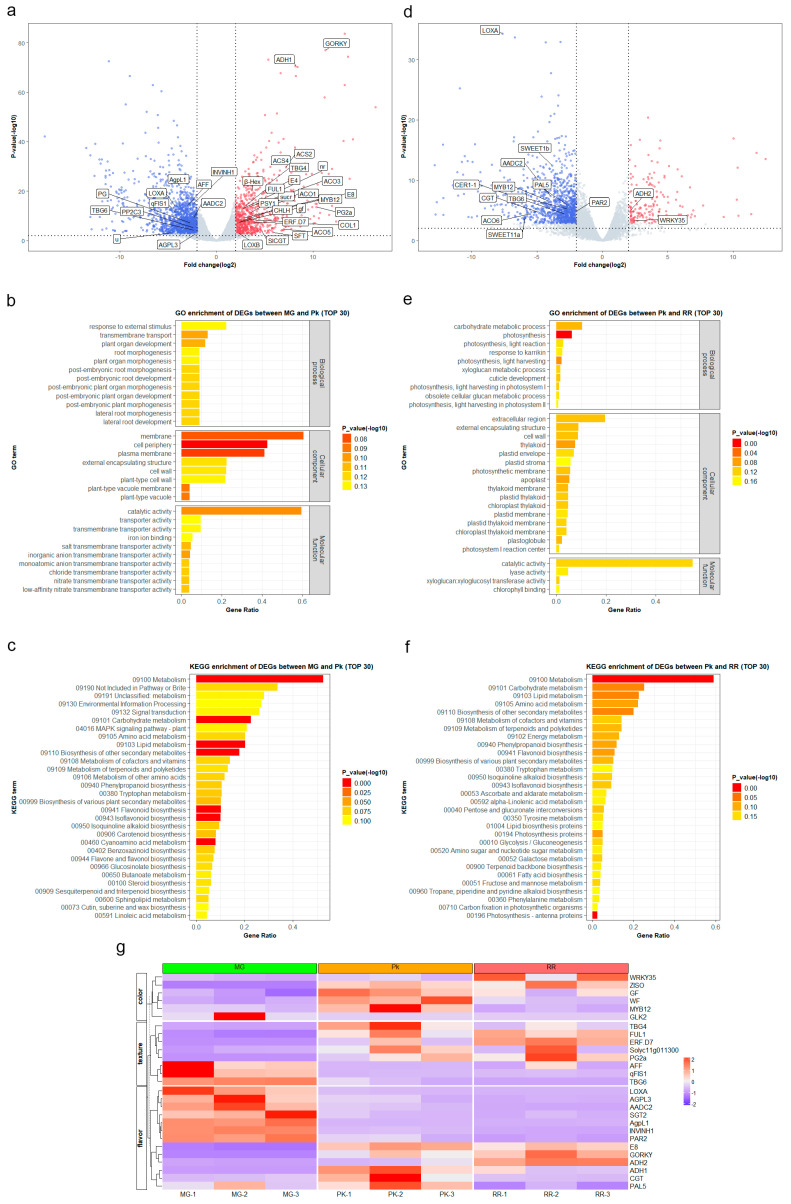
Transcriptome analysis of fruit at the mature green (MG), pink (Pk), and red ripe (RR) stages: (**a**) Volcano plot showing the differentially expressed genes (DEGs) between MG and Pk. Red dots represent upregulated genes, while blue dots represent downregulated genes. The same applies to (**d**). (**b**) GO enrichment analysis of DEGs between MG and Pk. (**c**) KEGG pathway enrichment analysis of DEGs between MG and Pk. (**d**) Volcano plot showing the DEGs between Pk and RR. (**e**) GO enrichment analysis of DEGs between Pk and RR. (**f**) KEGG pathway enrichment analysis of DEGs between Pk and RR. (**g**) Expression profiles of genes involved in color, texture, and flavor from MG to RR. The color transition from blue to red corresponds to increasing expression levels.

**Figure 3 plants-12-03505-f003:**
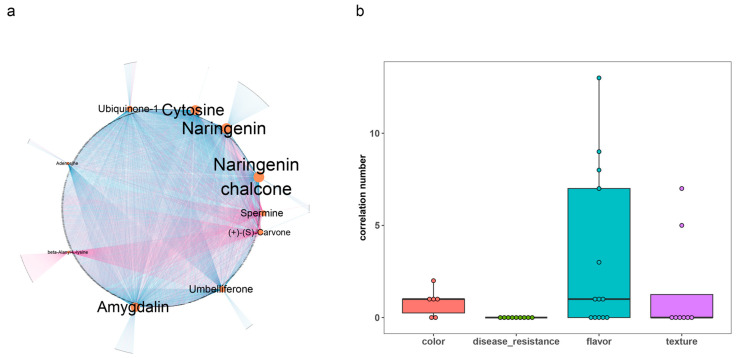
Correlation analysis between DAMs and DEGs: (**a**) Visualization of the correlation between DAMs (top ten) and DEGs in a network graph. Nodes represent DAMs and DEGs, and node size reflects the abundance of correlation. Correlated nodes are connected by edges, with pink indicating positive correlation and blue indicating negative correlation. (**b**) Box plot showing the abundance of correlated DAMs for well-characterized genes. Dots represent the well-characterized genes (color: *GF*, *SlWRKY35*, *UNIFORM*, *W*F, *SlMYB12*, and *ZISO*; disease resistance: *Cf-2*, *Cf-10*, *Ph-2*, *Sm*, *Rx4*, and four quantitative trait loci; flavor: *AADC2*, *ADH1*, *ADH2*, *AGPL1*, *AGPL3*, *E8*, *GORKY*, *INVINH1*, *LOXA*, *PAL5*, *PAR2*, *SGT2*, and *SlCGT*; and texture: *AFF*, *FUL1*, *PG2a*, *qFIS1*, *SlERF.D7*, *Solyc11g011300*, *TBG4* and *TBG6*).

**Figure 4 plants-12-03505-f004:**
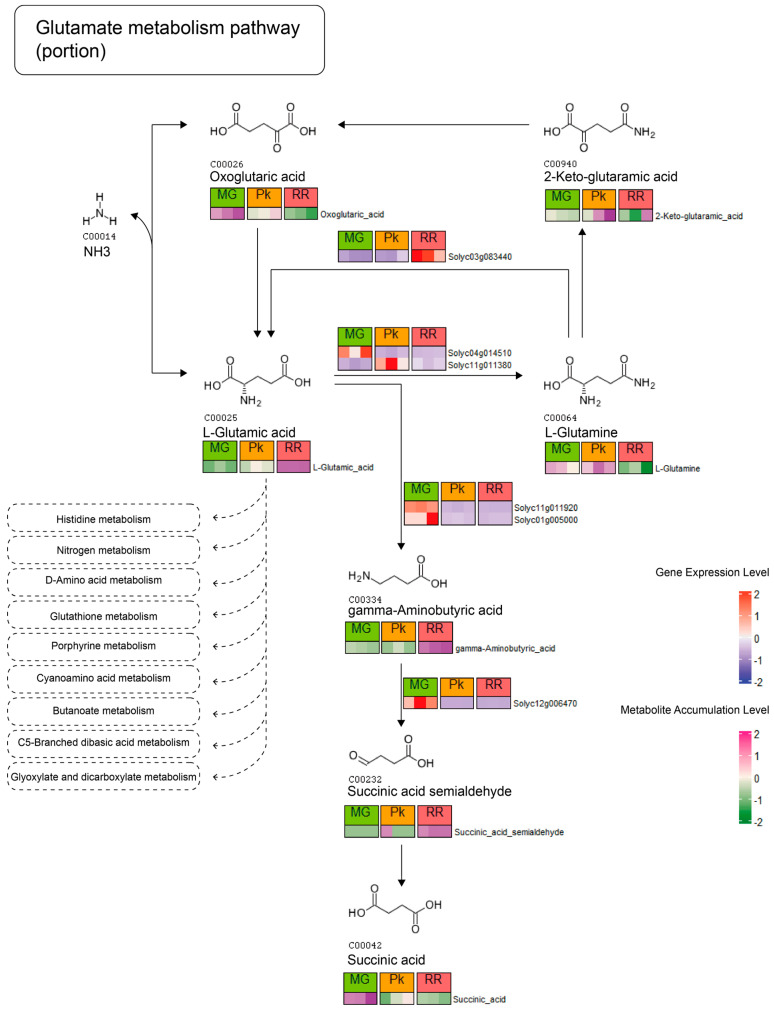
Glutamate metabolism pathway from KEGG, depicting the expression and accumulation profiles of DEGs and DAMs. The color gradient indicates the level of gene expression, ranging from blue (low expression) to red (high expression). Similarly, the color gradient represents the level of metabolite accumulation, transitioning from green (low accumulation) to pink (high accumulation).

**Figure 5 plants-12-03505-f005:**
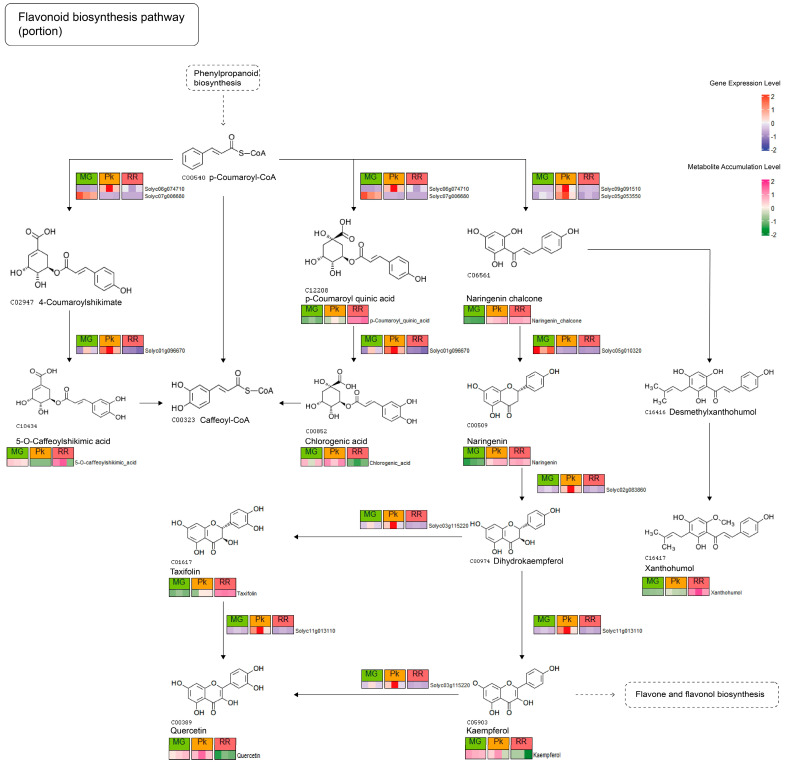
Flavonoid biosynthesis pathway from KEGG, illustrating the expression and accumulation profiles of DEGs and DAMs. The color gradient indicates the level of gene expression, ranging from blue (low expression) to red (high expression). Similarly, the color gradient represents the level of metabolite accumulation, transitioning from green (low accumulation) to pink (high accumulation).

## Data Availability

The genome sequencing datasets generated in this study have been deposited in the NCBI BioProject (https://www.ncbi.nlm.nih.gov/bioproject, accessed on 3 October 2023), accession number PRJNA1012661.

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
