# Peer review of "Transcriptome and Metabolome Provide Insights into Fruit Ripening of Cherry Tomato (Solanum lycopersicum var. cerasiforme)"

_plants, 2023, doi:10.3390/plants12193505_

Round 1

Reviewer 1 Report

The section materials and methods is too concise. For example, which is the method used to extract the metabolites? Did the authors do a single extraction and recover all kind of different metabolites? Which internal standard was used? How many mg of fruit material were used to prepare the sample? It is said that the complete fruit was frozen. Did the authors used the complete fruit including the jelly and seed material?

For RNA samples again how many mg fruit material was used? Which is the method used for isolating RNA and mRNA for constructing the libraries?

Line 86: it is said that 25 nucleic acid metabolites were identified. It is not correct to name those metabolites nucleic acid because a nucleic acid is a polimer of nucleotides either of different kind. What you have measured are nucleotides or nucleotide derivatives so please rename this group of compound all along the manuscript.

Please Add in table S2 which compunds are included in each subclass in the metabolite set

The Figure numbering starts with Figure 1b. It is not mentioned in the text Figure 1a. Please include a sentence to refer to Fig 1 a which I believe should be included in the manuscript.

Line 89: 50 metabolites increasing but in the figure (Venn diagram Figure 1b) seems to be only 41. Please check

Line 96-97 please check the numbers which are not the same as presented in the figure 1B (venn diagram)  45 up and 33 down in the diagram vs 39 up and 40 down in the text.

Please add in table S3 and S4 (by adding a column in the provided table) the set of metabolite to which each metabolite belong to help the reader to know which are the metabolites belonging for example to benzamine, indol, etc sets.

Line 103-112: please check the numbers of genes increasing and decreasing as those presented in the text does not fit the venn diagram.

Line 149: form the GO annotations it is not possible to say that those classes suggest the cessation of photosynthesis and the formation of flavor. Because you are not saying what happens with the genes related to those processes here. Please rephrase the idea.

Line155: you mention you used well known genes involved in color texture and flavor were analyzed. Please add a supplementary table including all these genes and the references where they appeared characterized in relation to what you want to show.

Line 160: replace exhibited with showed lower expression at the Pk and RR stage when compared to MG stage

Line 164: For example, the expression of ADH2,

Line 165-166: replace exhibited with was continuously upregulated from….

Line 168: you mention 80 commen KEGG terms please mark those in bold in the corresponding tables S8 and S11

Also dering those KEGGS specific for MG-Pk and Pk-RR transitions please a color code and use bold to mark them in the S8 and S11 tables respectively

Please add which genes related to desease resistance you are considering in figure 3b. You could add this list in the same suplementary table from Line 155 (see above)

For the integrative analysis Line 217 please you have to name each gene (what are the genes encoding for what enzyme) in the text and in the figures. Refer and compare to works previously done in tomato

Line 272: you mention glutamic which was already analyzed and explained before but phenol is the first time you mention as related to flavor please explain in which extent is acting phenol in tomato flavor

Line 272 -274: you do not find general flavor compounds such as sugars and citric acid in your work. I suppose that this is the result of using a complex matrix such as the complete fruit but not pericarp tissue as usually done. If the metabolite levels are already stable prior MG stage this you could easily measure in fruits younger than MG stage. So Please discuss considering the genes involved in the metabolic pathways of those metabolites and add data to validate the assumption you made.

Line 288: replace since with beginning at 

Line 302: in figure 5 does not appear dattelic acid. Could you please add this metabolic pathway?

Author Response

Comment 1: The section materials and methods is too concise. For example, which is the method used to extract the metabolites? Did the authors do a single extraction and recover all kind of different metabolites? Which internal standard was used? How many mg of fruit material were used to prepare the sample? It is said that the complete fruit was frozen. Did the authors used the complete fruit including the jelly and seed material?

[Finish]Response 1: We apologize for not including the specific details of our materials and methods in the manuscript. We have now added a detailed description in the manuscript that explicitly outlines our experimental procedures, ensuring that readers can accurately understand the details of our study. These sentences are highlight in green.

Comment 2: For RNA samples again how many mg fruit material was used? Which is the method used for isolating RNA and mRNA for constructing the libraries?

[Finish]Response 2: As above, we have added a detailed description in the manuscript. These sentences are highlight in green.

Comment 3: Line 86: it is said that 25 nucleic acid metabolites were identified. It is not correct to name those metabolites nucleic acid because a nucleic acid is a polimer of nucleotides either of different kind. What you have measured are nucleotides or nucleotide derivatives so please rename this group of compounds all along the manuscript.

[Finish]Response 3: Thanks for your suggestion. The naming of the group was provided by the software MetaboAnalyst 5.0, the classification was based on RefMet Classification (https://www.metabolomicsworkbench.org/databases/refmet/refmet_classification.php). Nucleic acids, as a super-class, encompass various major-classes including dinucleotides, flavins, glycinamide ribonucleotides, imidazoles, nicotinamides, purines, pyrimidines, and so on.

To avoid any misunderstanding among readers, we have decided to consider your suggestion.We have renamed this group of compounds as your suggestion all along the manuscript and supplementary files, and we have redrawn the pie charts in FigureS1. The revised sentences are highlight in yellow.

Comment 4: Please Add in table S2 which compounds are included in each subclass in the metabolite set 

[Finish]Response 4: Thanks for your suggestion. We have added the compounds in TableS2.

Comment 5: The Figure numbering starts with Figure 1b. It is not mentioned in the text Figure 1a. Please include a sentence to refer to Fig 1 a which I believe should be included in the manuscript.

[Finish]Response 5: Thank you for your suggestion. We agree that it was inappropriate to mention Figure1a in the Introduction section. We have now added a description of Figure1a in the Results section as you recommended. The revised sentences are highlight in yellow.

Comment 6: Line 89: 50 metabolites increasing but in the figure (Venn diagram Figure 1b) seems to be only 41. Please check

[Finish]Response 6: Thank you for your scrupulous review. We sincerely apologize for our carelessness in submitting an incorrect Venn diagram. We have now resubmitted the correct Venn diagram (Figure1b). Actually, the number of increasing metabolites is 50 (TableS3), hence, the textual content in the manuscript remains unchanged.

Comment 7: Line 96-97 please check the numbers which are not the same as presented in the figure 1B (venn diagram) 45 up and 33 down in the diagram vs 39 up and 40 down in the text.

[Finish]Response 7: Thanks for your scrupulous review. The correct Venn diagram was resubmitted (Figure1b) and the textual content in the manuscript remains unchanged.

Comment 8: Please add in table S3 and S4 (by adding a column in the provided table) the set of metabolite to which each metabolite belong to help the reader to know which are the metabolites belonging for example to benzamine, indol, etc sets.

[Finish]Response 10: Thank you for your insightful suggestion. We have now added the super class of each metabolite in Table S3 and Table S4, as you recommended. This will provide clarity and enhance the understanding of our data.

Comment 9: Line 103-112: please check the numbers of genes increasing and decreasing as those presented in the text does not fit the venn diagram.

[Finish]Response 9: Thanks for your scrupulous review. The correct Venn diagram was resubmitted (Figure1b) and the textual content in the manuscript remains unchanged.

Comment 10: Line 149: form the GO annotations it is not possible to say that those classes suggest the cessation of photosynthesis and the formation of flavor. Because you are not saying what happens with the genes related to those processes here. Please rephrase the idea.

[Finish]Response 10: Thanks for your insightful suggestion. We have realized that such conclusions may be hasty so we deleted these statements about GO annotation and presented this part as a simple result.

Comment 11: Line155: you mention you used well known genes involved in color texture and flavor were analyzed. Please add a supplementary table including all these genes and the references where they appeared characterized in relation to what you want to show.

[Finish]Response 11: Thanks for your practical suggestion. We were remiss in not providing information about these genes. We have added the information of these genes in TableS13, and mentioned in manuscript. The revised sentences are highlight in yellow.

Comment 12: Line 160: replace exhibited with showed lower expression at the Pk and RR stage when compared to MG stage

[Finish]Response 12: Thank you for your suggestion. We have revised the sentence and highlight it in yellow.

Comment 13: Line 164: For example, the expression of ADH2,

Comment 14: Line 165-166: replace exhibited with was continuously upregulated from….

[Finish]Response 13-14: Thank you for your alternative suggestions. We have revised the sentence as you suggested in Comment 14. The revised sentences are highlight in yellow.

Comment 15: Line 168: you mention 80 commen KEGG terms please mark those in bold in the corresponding tables S8 and S11

Comment 16: Also dering those KEGGS specific for MG-Pk and Pk-RR transitions please a color code and use bold to mark them in the S8 and S11 tables respectively

[Finish]Response 15-16: Thanks for your practical suggestions. We have marked these terms and added annotation information in TableS8 and TableS11.

Comment 17: Please add which genes related to desease resistance you are considering in figure 3b. You could add this list in the same suplementary table from Line 155 (see above)

[Finish]Response 17: Thanks for your suggestions. We have added the gene information in the footer of Figure3. And the information of these genes were provided in TableS13.

Comment 18: For the integrative analysis Line 217 please you have to name each gene (what are the genes encoding for what enzyme) in the text and in the figures. Refer and compare to works previously done in tomato

[Finish]Response 18: Thanks for your suggestions. We have added the encoding information of genes in the manuscript text. The encoding information was referred to NCBI gene database. The revised sentences are highlight in yellow.

Though some genes possess comprehensible gene symbol (such as Solyc11g011380: GS1), there’s still some genes possess only symbol code (such as Solyc03g083440: LOC101254281). This will lead to non-uniform symbols in the figures. In order to ensure the uniformity of the gene symbol, we would like to preserve the gene ID “Solyc…” in the figures. To enhance the readability of the Figure4 and Figure5, we added a TableS14 in the supplementary file, providing the gene information.

Comment 19: Line 272: you mention glutamic which was already analyzed and explained before but phenol is the first time you mention as related to flavor please explain in which extent is acting phenol in tomato flavor

[Finish]Response 19: Thanks for making us aware of the logical inadequacy of our writing.

Admittedly, we mentioned the phenol in the identification of the metabolites, as aromatic metabolites might contribute to fruit flavor. However, when we performed the integrative analysis on KEGG metabolic pathway, we identified more DEGs and DAMs in the glutamate metabolism pathway and flavonoid biosynthesis pathway, facilitating our analysis on this pathway. Perhaps research on volatile aromatic metabolites is largely unknown, it is difficult to discuss these pathways through transcriptome and metabolome information.

Hence, to avoid the possible logical confusion, we revised the sentences as “Previous studies have demonstrated a significant association between glutamic acid and overall fruit flavor intensity and consumer preference [41]. Meanwhile, more DAMs and DEGs were identified in glutamate metabolism pathway, facilitating us to gain insights into the metabolic transformations in this pathway (TableS14)”.

Comment 20: Line 272 -274: you do not find general flavor compounds such as sugars and citric acid in your work. I suppose that this is the result of using a complex matrix such as the complete fruit but not pericarp tissue as usually done. If the metabolite levels are already stable prior MG stage this you could easily measure in fruits younger than MG stage. So Please discuss considering the genes involved in the metabolic pathways of those metabolites and add data to validate the assumption you made.

[Finish]Response 20: Thanks for your insightful suggestion. As you so wisely surmised, we used the mixed samples, including all of the fruit. We analyzed the genes involved in the metabolites according to your suggestion. And we observed the expression levels of these traits-associated gene loci identified through GWAS [Tieman et al, Science, 2017]. Among the 41 genes, only 2 were found to be included in our identified DEG list, while the expression levels of the remaining genes did not show significant changes during the MG-RR stages (TableS15). These data might support our assumption These data may preliminarily, and further investigations involving a larger population are anticipated in the future. The revised sentences are highlight in yellow. We appreciate your insightful suggestions on our manuscript!

Comment 21: Line 288: replace since with beginning at 

[Finish]Response 21: Thanks for your suggestions. We have replaced the “since” with “beginning at”. The revised sentences are highlight in yellow.

Comment 22: Line 302: in figure 5 does not appear dattelic acid. Could you please add this metabolic pathway?

[Finish]Response 22: Thanks for your carefully review. We are sorry that our oversight has led to this confusion. Actually, this metabolic pathway is in the figure 5. “Dattelic acid” is the compound name in the BIODEEP database, while in the KEGG COMPOUND database, its ID is C10434 and the compound name is “5-O-caffeoylshikimic acid”. We apologize for the inconsistency in the usage of compound names in both the text and images, which caused this confusion. To avoid such errors, we have now uniformly changed the compound name to 5-O-caffeoylshikimic acid throughout the entire manuscript and made corresponding modifications to the figures and tables.

Reviewer 2 Report

The manuscript describes differentially accumulated metabolites and genes related to flavor during fruit ripening of tomato cherry. The paper is well-written, describes a good analysis of results, and has an adequate discussion. However, some details need to be improved before publication.
Line 23, support instead "sup-port"
 Please add the full name of the abbreviated metabolites and genes in the main text. In addition, all genes must be in italics. For example, see line 51.
More detailed methodology should be included, for example, the number of fruits per stage used for each extraction, quantity of fruit for RNA and metabolite extraction and number of replicates, RNA extraction kit or method, type of solvent (methanol 100%)

Author Response

Comment 1: Line 23, support instead "sup-port"

[Finish]Response 1: Thanks for your suggestions. We have replaced the “sup-port” with “support”.

Comment 2: Please add the full name of the abbreviated metabolites and genes in the main text. In addition, all genes must be in italics. For example, see line 51.

[Finish]Response 2: Thanks for your scrupulous review. We have reviewed the full text and followed your suggestion.

Comment 3: More detailed methodology should be included, for example, the number of fruits per stage used for each extraction, quantity of fruit for RNA and metabolite extraction and number of replicates, RNA extraction kit or method, type of solvent (methanol 100%)

[Finish]Response 3: We apologize for not including the specific details of our materials and methods in the manuscript. We have now added a detailed description in the manuscript that explicitly outlines our experimental procedures, ensuring that readers can accurately understand the details of our study. These sentences are highlight in green.

Reviewer 3 Report

Introduction

A very brief overview of the situation of cherry tomatoes in the world would be welcome: spread, diversity, importance.

L 34-36. Some figures, percentages, comparing cherry tomatoes and large-fruited tomatoes, related to the fruit content in sugar, organic compounds and volatile compounds are necessary. The description would be more suggestive for the readers.

L 37-38. The stages described for ripening cherry tomatoes are only valid for varieties which at full ripeness are red in colour. There are many varieties which at maturity are yellow or greenish with orange stripped, reddish black, etc. Let it be clearly stated to which varieties of cherry tomatoes they refer.

L 47-49. Under what circumstances are these bitter compounds formed? Please state clearly!

Results

L 206-210. The authors make correlations between differentially accumulated metabolites (DAM) and disease resistance genes, but no reference to tomato diseases is made in the introductory part. On the other hand, it would be preferable if the issue of tomato diseases were not treated in general but separately, since the diversity and mode of action of pathogens is very different : viruses, bacteria, fungi, local infection, systemic infection, etc. 

Discussions

L 301-307. Although the paper mentions dattelic acid and chlorogenic acid as having a particular role in human health, the authors make no mention of lycopene - a carotenoid of prime importance in the control and especially prevention of cancer. 

L 308-309. The paper states that "The long-term improvements made in tomato fruit size, firmness, and disease re-308 sistance have led to a deterioration in fruit flavor [1,41]". Should it be made clear in what sense these characteristics influence fruit flavour?

Materials and working method

The authors should give some details on the cultivation technology applied to tomatoes as fertilisation, irrigation, light, etc. are essential for fruit quality in general and flavour in particular.

Author Response

Introduction

Comment 1: A very brief overview of the situation of cherry tomatoes in the world would be welcome: spread, diversity, importance.

[Finish]Response 1: Thanks for your suggestions. We have added the information in the Introduction section. The revised sentences are highlight in blue.

Comment 2: L 34-36. Some figures, percentages, comparing cherry tomatoes and large-fruited tomatoes, related to the fruit content in sugar, organic compounds and volatile compounds are necessary. The description would be more suggestive for the readers.

[Finish]Response 2: Thanks for your suggestions. We are agreed with you that figures and percentages would be more suggestive for the readers. Previous researches have provided the compare between the cherry tomatoes and “big-fruit tomatoes” (Hobson & Bedford, Journal of Horticultural Science, 1989), we cited it and performed a simple calculation. However, volatile compounds are various, Causse (Book: Fruit and Vegetable Flavour, 2008) had reviewed the distinctions though the research (Tikunov et al., Plant Physiology, 2005), we cited his conclusion directly. The revised sentences are highlight in blue.

Comment 3: L 37-38. The stages described for ripening cherry tomatoes are only valid for varieties which at full ripeness are red in colour. There are many varieties which at maturity are yellow or greenish with orange stripped, reddish black, etc. Let it be clearly stated to which varieties of cherry tomatoes they refer.

[Finish]Response 3: Thanks for your scrupulous review. We are agreed with you. We have limited the scope of the statement to make it more precise. The revised sentences are highlight in blue.

Comment 4: L 47-49. Under what circumstances are these bitter compounds formed? Please state clearly!

[Finish]Response 4: Thank you very much for reminding. Our statement here may be ambiguous. Actually, what we're trying to state is that metabolites can affect flavor, aiming to suggest that metabolome analysis can be used to understand flavor formation. So, we revised this sentence to avoid unnecessary implication. The revised sentences are highlight in blue.

Results

Comment 5: L 206-210. The authors make correlations between differentially accumulated metabolites (DAM) and disease resistance genes, but no reference to tomato diseases is made in the introductory part. On the other hand, it would be preferable if the issue of tomato diseases were not treated in general but separately, since the diversity and mode of action of pathogens is very different : viruses, bacteria, fungi, local infection, systemic infection, etc. 

[Finish]Response 5: Thank you very much for reminding. Our statement here may be ambiguous. Previous studies mentioned that the long-term improvements made in tomato fruit size, firmness, and disease resistance have led to a deterioration in fruit flavor. Here, we attempt to explain the correlation between gene expression and metabolite accumulation, rather than the effect of disease treatment on metabolites. Meanwhile, the introduction section introduces an overview to the ripening process and flavor formation process of fruit, so there is no reference to tomato diseases.

So, to enhance the readability of this section, we added a sentence to state our purpose at the beginning of this paragraph. The revised sentences are highlight in blue.

Discussions

Comment 6: L 301-307. Although the paper mentions dattelic acid and chlorogenic acid as having a particular role in human health, the authors make no mention of lycopene - a carotenoid of prime importance in the control and especially prevention of cancer. 

[Finish]Response 6: Thank you very much for reminding. This problem stimulated our discussion. We agree that lycopene is an important compound of tomato. However, lycopene was not identified by our metabolite identification, neither was identified in previous study (Zhu et al, Cell, 2018). As we know, the lycopene is insoluble in methanol, while the methanol a necessary solvent for the extraction of metabolites in UHPLC-MS/MS. That is to say, the lycopene haven’t be identified by UHPLC-MS/MS method. Hence, it is difficult to discuss an important compound that has not been detected.

Comment 7: L 308-309. The paper states that "The long-term improvements made in tomato fruit size, firmness, and disease re-308 sistance have led to a deterioration in fruit flavor [1,41]". Should it be made clear in what sense these characteristics influence fruit flavour?

[Finish]Response 7: Thank you very much for reminding. Previous study has partly explained this issue (Zhu et al, Cell, 2018). Introgression segments from wild relatives might bring the genes affecting flavor and led to large-scale change of metabolites. Our result suggested that selection for firmness-related genes might have directly led to a greater deterioration of fruit flavor compared to the selection for disease resistance genes, thus enhancing our understanding of the impact of the breeding improvement process on flavor. So we revised this paragraph to highlight the sense of our study.

Materials and working method

Comment 8: The authors should give some details on the cultivation technology applied to tomatoes as fertilisation, irrigation, light, etc. are essential for fruit quality in general and flavour in particular.

[Finish]Response 8: Thanks for your practical suggestions. We have added a detailed description in the manuscript about the cultivation technology. These sentences are highlight in green.

Round 2

Reviewer 1 Report

Line 178-182: please revise gramar

Line 245: revire PK-MG stage, shouldn’t be PK-RR or MG-PK ??

Line 178-182: please revise grammar

Author Response

Comment 1: Line 178-182: please revise grammar

[Finish]Response 1: We have made modifications to the entire paragraph to ensure compliance with English grammar rules. The revised portions are highlighted in pink.

Comment 2: Line 245: revire PK-MG stage, shouldn’t be PK-RR or MG-PK ??

[Finish]Response 2: Thanks for your scrupulous review. It should be written as Pk-RR here. We have checked the entire manuscript to ensure that no more writing errors occur due to carelessness. The revision has been marked in pink.

Reviewer 3 Report

Congratulations on your hard work in writing this paper!!!

Author Response

Congratulations on your hard work in writing this paper!!!

[Finish]Response: Thank you for investing your valuable time and effort in providing us with valuable comments and suggestions on our manuscript! Your suggestions have contributed to making our manuscript more rigorous and complete!